# Early Triassic super-greenhouse climate driven by vegetation collapse

Zhen Xu [1,2] ✉, Jianxin Yu [1] ✉, Hongfu Yin[1], Andrew S. Merdith[3], Jason Hilton [4], Bethany J. Allen [5,6], Khushboo Gurung [2], Paul B. Wignall[2], Alexander M. Dunhill [2], Jun Shen [1], David Schwartzman[7], Yves Goddéris [8], Yannick Donnadieu [9], Yuxuan Wang[1,2], Yinggang Zhang [2], Simon W. Poulton [2,10] & Benjamin J. W. Mills [2] ✉

The Permian–Triassic Mass Extinction (PTME), the most severe crisis of the Phanerozoic, has been attributed to intense global warming triggered by Siberian Traps volcanism. However, it remains unclear why super-greenhouse conditions persisted for around five million years after the volcanic episode, with one possibility being that the slow recovery of plants limited carbon sequestration. Here we use fossil occurrences and lithological indicators of climate to reconstruct spatio-temporal maps of plant productivity changes through the PTME and employ climate-biogeochemical modelling to investigate the Early Triassic super-greenhouse. Our reconstructions show that terrestrial vegetation loss during the PTME, especially in tropical regions, resulted in an Earth system with low levels of organic carbon sequestration and restricted chemical weathering, resulting in prolonged high $CO_2$ levels. These results support the idea that thresholds exist in the climate-carbon system whereby warming can be amplified by vegetation collapse.

The latest Permian to Early Triassic (~252–247 million years ago; Ma) was a period of intense environmental and biotic stress[1,2]. During the Permian–Triassic Mass Extinction (PTME) at ~252 Ma, around 81–94% of marine invertebrate species and 89% of terrestrial tetrapod genera became extinct[2]. It is generally agreed that the PTME was driven by volcanogenic carbon emissions from Siberian Traps volcanism, potentially coupled with additional thermogenic releases, resulting in intense greenhouse warming[3–10]. A major negative excursion in carbonate $\delta^{13}C$ ratios, over a time interval of about 50–500 thousand years (kyrs), supports the notion of a major carbon cycle perturbation[4–6]. However, it is not well understood why the extreme hothouse climate persisted throughout the 5 million years (Myrs) of

the Early Triassic. The precise time interval of Siberian Traps degassing is uncertain, although the main phase of volcanism occurred around the Permian-Triassic Boundary (PTB), possibly with a further pulse about two million years later, during the Smithian Substage of the Early Triassic[8]. Nevertheless, it would normally be expected that atmospheric $CO_2$ and global surface temperature should have declined to pre-volcanism levels within ~100 kyr of the volcanic pulses, due to amplified global silicate weathering and/or increased burial of organic carbon[2].

The unusual multimillion-year persistence of super-greenhouse conditions has sparked considerable debate, and it has been suggested that it may be linked to a change in the silicate weathering feedback,

[1]State Key Laboratory of Geomicrobiology and Environmental Changes, School of Earth Sciences, China University of Geosciences, Wuhan 430074, P.R. China. [2]School of Earth and Environment, University of Leeds, Leeds, UK. [3]School of Physics, Chemistry and Earth Science, University of Adelaide, Adelaide, SA, Australia. [4]Birmingham Institute of Forest Research, University of Birmingham, Edgbaston, Birmingham, UK. [5]Department of Biosystems Science and Engineering, ETH Zürich, Basel, Switzerland. [6]Computational Evolution Group, Swiss Institute of Bioinformatics, Lausanne, Switzerland. [7]Department of Biology, Howard University, Washington, DC, USA. [8]Géosciences Environnement Toulouse, CNRS-Université de Toulouse III, Toulouse, France. [9]CEREGE, Aix Marseille Université, CNRS, IRD, INRA, Coll France, Aix-en-Provence, France. [10]State Key Laboratory of Geological Processes and Mineral Resources, China University of Geosciences, Wuhan 430074, P.R. China. ✉e-mail: Z.xu@leeds.ac.uk; yujianxin@cug.edu.cn; B.Mills@leeds.ac.uk

such that $CO_2$ could not be efficiently removed from the surface system[11]. This could potentially have been due to reduced availability of weatherable material from erosion[11], which would limit global silicate weathering rates[12,13]. Alternatively, continental weathering may have been rapid and accompanied by high rates of reverse weathering in a silica-rich ocean, removing silicate mineral-derived cations into clays instead of forming carbonate minerals, and thus limiting overall $CO_2$ drawdown[14,15]. These are intriguing hypotheses, but it remains unclear why a severe reduction in global erosion, and/or an episode of high ocean silica levels, would necessarily persist for ~5 Myrs and then recover during the Middle Triassic. Although uncertain, existing compilations of sedimentation rates[16,17] and the maintenance of sporadic siliceous rock records across and after the PTME[18] (see Supplementary Fig. S1) are not clearly supportive of these timings, and suggest that while these processes likely contributed to climate regulation, our understanding of the timeframe of super-greenhouse conditions remains incomplete.

Here, we explore a further mechanism for elevated Early Triassic temperatures that is closely tied to the timeframe of extreme warmth. This approach is based on the concept of an 'upper temperature steady state', in which a change in the Earth system caused the climate-carbon cycle to stabilize at a much higher global temperature for millions of years[19]. Specifically, we investigate the hypothesis that the key driver of the transition to a super greenhouse Earth was the dramatic and prolonged reduction of low-latitude terrestrial biomass caused by the PTME[20,21] and its delayed recovery[22]. Tropical peat-forming ecosystems are responsible for substantial drawdown of $CO_2$, but these extensive biomes were lost at the end of the Permian[20,23–25]. Indeed, plant species richness and abundance dropped significantly during the Permian–Triassic transition, which is the only genuine mass extinction level event of land plants through the whole Phanerzoic[26], leaving a multimillion year "coal gap" in the Early to Middle Triassic where terrestrial plant materials did not build up as peat[20,23,25]. To test this hypothesis, we quantify the distribution of terrestrial plant productivity across the PTME and Early-to-Middle Triassic from the plant fossil record and use this information to guide a linked climate-biogeochemical model of the Early Triassic hothouse, testing whether these biotic changes may have resulted in a higher temperature steady state.

## Reconstructing plant biogeography across the PTME and Early-Middle Triassic

Our plant fossil database, including macrofossil and palynology data from the latest Permian to the Middle Triassic, is summarized in Fig. 1 and further detailed in Supplementary Tables S1–S3. Non-marine chronostratigraphy comes from recently published data (Fig. S3, "Methods" 1), with lithological, sedimentary features, and clastic strata thickness in each basin evaluated for the influence of taphonomy on plant fossil and biomass preservation (see SI text 1 for details). Considering the correlation resolution achievable for terrestrial strata, this study uses a stage-level resolution as used in previous studies on the Permian-Triassic transition[20]. Having correlated localities using carbon isotope stratigraphy and mercury peaks, we show, using plant biomarker data from many localities, the collapse of terrestrial floras occurred around the Permian-Triassic Boundary, with most losses in the latest Permian[27,28]. Statistical methods, including Squares and interpolation diversity testing, were used to evaluate the influence of fossil sampling intensity, and demonstrate that our approach is robust to variation in fossil density at the global scale (see "Materials and Methods" 4 for details) (Fig. 2).

As fossil plants are typically fragmented prior to fossilization, all plant fossil records were normalized[20,29] to reduce artefacts of palaeobotanical nomenclature (see "Methods" 2 for details). Normalization compensates for the palaeobotanical practice of assigning different plant organs (e.g., roots, stems, leaves, cones and seeds) of the same plant to separate fossil genera and species[20]. We selected a plant organ whose fossil taxonomy is most likely to reflect the whole plant taxonomy and omitted other organs that belong to the same plant group, to avoid duplication[29]. As an example, normalization removed ~20% of the South China Changhsingian macro plant species as duplicates[20]. We identified parent plants of plant microfossil (spore and pollen) data where known, to supplement plant macrofossil records. Plant macrofossils mostly recorded the lowland vegetation landscape, while plant micro fossils also record upland plant species richness information[20,29,30].

Diversity estimates and inferences from plant morphological traits were used to construct climate-linked plant biomes, then this information was collectively used to extrapolate biomes across corresponding climate zones (see "Methods" 5 for details). The analysis of plant character and function across palaeogeographic regions involved three steps. First, plant morphological traits related to physiological functions were extracted from plant fossils. Among all plant functional traits, whole plant height and shape, position in the flora, leaf size, vein type, vein density and relative cuticle thickness, which are related to plant productivity, biomass and water requirements/resistance to drought, were measured or semi-quantitatively estimated in late Permian to Middle Triassic plant fossils (Table S5)[31,32]. Floras were assigned using the Köppen-Geiger climate classification system according to their habitat information within the plant functional traits (see "Methods" 5 for details, Table S6). For example, gigantopterids are assigned to the rainforest group with giant leaves, 'drip tips' and intricate vein networks, indicating their high moisture requirements and high efficiency of carbon and nutrient transport, similar to recent angiosperm dominant rainforest[20,31,33,34]. The Cathaysian flora with a high proportion of gigantopterids is of high spatial complexity, including a canopy of tall *Lepidodendron* lycopod trees, diverse understory tree ferns and sphenophytes, and gigantopterids and ferns, supporting the presence of widespread late Permian rainforests in the South China Lowland (see Supplementary Table S5, S6). Secondly, the floristic information from the known floras and fossil plants was assigned to the less known floras by comparing the similarity in plant taxon composition (see "Methods" 6 for details). The floral comparison is partly based on macrofossil family level clustering, and partly on the species richness in each morphological category. Thirdly, we expanded our reconstructed plant distributions beyond the fossil evidence by assuming they would colonize any regions of tolerable climate—aiming to capture 'hidden' communities such as the upland gymnosperms recorded in the palynological record[20,25,29,35] (see "Methods" 7 for details). Plant fossil records combined with lithological indicators of local climate (e.g., coals, evaporites, tillites), were transferred onto a palaeogeographic grid map with a resolution of $40 \times 48$ (Fig. 3). These local (i.e., per grid box) data were then used collectively to extrapolate biomes across corresponding climate zones. Terrestrial tetrapod fossil occurrences served as an indicator for the existence of vegetation to aid in extrapolation, whereas lithological indicators of aridity are used to prevent extrapolation into desert regions (see "Methods" 8 for details) (Fig. 3).

Figures 1 and 2 highlight the more substantial extinction of low−middle latitude (−45°N−45°S) tropical−subtropical vegetation, especially lowland forests, during the PTME, with 86% macrofossil species extinction in low−middle latitudes, as opposed to 66% in high latitudes (see Table S5). The published local sedimentological and lithological surveys from Siberia, Xinjiang, NW China, SW China, Utah, western Europe, South Africa, Australia, Antarctica, and Argentina, spanning a broad spectrum of latitudes from north to south, show that the 'coal gap' after the PTME was not associated with a significant loss of river or delta sediments in these areas (Figure S3, see "Methods" 9 for details). The existence of low diversity pioneer floras in South China indicates that the preservation window was not closed even in some Early Triassic low-latitude areas with the highest post-PTME

temperatures and extinction magnitude[20]. Therefore, it appears that the removal of vegetation (especially lowland plants), rather than taphonomy, was likely the main cause of the low plant abundance, low sedimentary organic carbon contents, and general lack of other plant-related chemicals such as biomarkers in sediments during the Early Triassic[20]. Before the PTME, plant macrofossil species richness was greatest in low–mid latitude areas, while after the crisis, high latitude richness was much higher (Figs. 1B, 2). Although we compiled all published fossil data known to us, and investigated the sedimentary facies and thickness of documented sections to minimize the influence of taphonomic bias, the complexity of the Earth system remains challenging to fully reconstruct. Nevertheless, we believe that this study offers valuable approaches to addressing this issue and represents a significant step toward improving our understanding of the spatio-temporal distribution of flora during the PTME and its aftermath.

The reversal of the modern latitudinal diversity gradient is also seen in terrestrial tetrapods after the PTME[36], suggesting that this biogeographic transition may have been ubiquitous across ecosystems on land. The tetrapod "Dead Zone" between 30°N and 40°S may reflect the extinction of terrestrial primary producers in low–middle latitude

lowland areas with limited upland survivors[37] (Figs. 1B, 2). The latitudinally symmetric pattern of the terrestrial biosphere suggests that the primary extinction mechanism had a similarly distributed spatial impact. Evidence suggest that the various potential factors related to volcanism, including acid rain, heavy metals, toxic gases, UVB radiation, and climate change, may have possibly contributed to the terrestrial extinction[2,9,24,38–41]. Among these, climate change induced by LIP activity stands out for its global and latitudinal effects[22]. Application of the HadCM3 climate model suggests that extreme climatic consequences—such as El Niño-driven intensified heat stress and seasonal aridity—were prevalent in low- to mid-latitude regions[9]. These regions notably lack plant and tetrapod fossils after the PTME, suggesting that these climate changes likely served as a primary driver of terrestrial extinctions[9].

Figure 1A and S2 show that the global terrestrial palaeophytogeographical feature of the Permian–Triassic interval is the replacement of the low-latitude tropical Cathaysian flora, the low–middle latitude temperate–subtropical Euramerica flora, the high-latitude boreal Angara flora, meridional Gondwana flora, and mixed floras, by a uniform herbaceous lycopod-dominated flora in the Early Triassic, in general accord with previous studies with more limited global

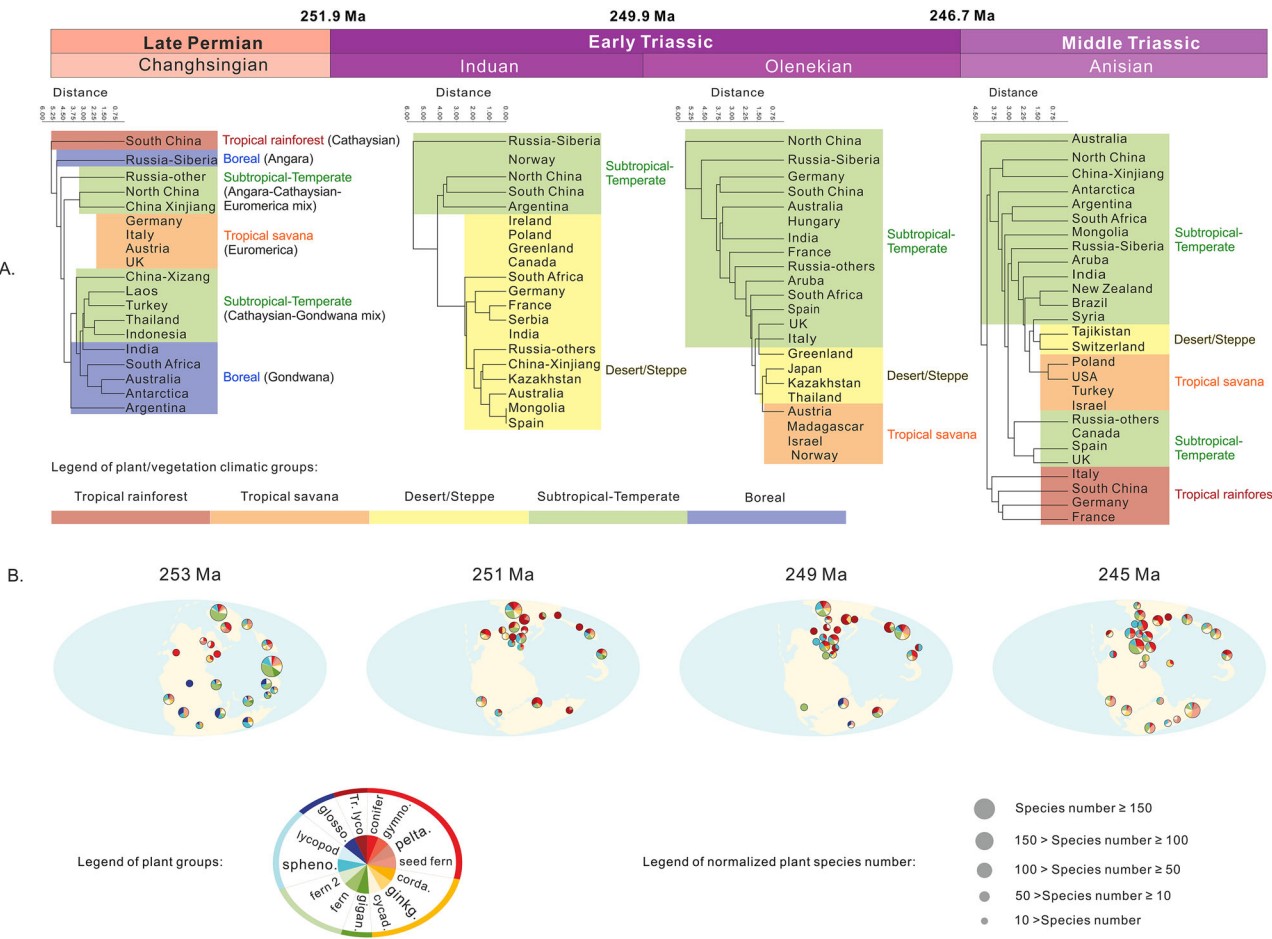

**Fig. 1 | Late Permian to Middle Triassic plant family level clustering, morphological categories and species richness by latitude.** Full data in Supplementary Tables S1, S2, S3 and S4. All data used in this figure are normalized for fragmentation (see text). **A** Trees show clustering of flora in each area by plant family composition, with the corresponding climate zone abbreviation listed on the branches. The climate zones are highlighted by color bar. The name of the late Permian Changhsingian climatic group from previous studies is listed after the climate zones in brackets. Areas lacking macro plant fossil records do not have associated branches and are classified using palynological data. **B** Floras indicated by plant macrofossils, microfossils and tetrapod fossils. The small pie charts represent the floras studied, showing the plant composition, with the number of species shown by the size of the pie chart. Legend abbreviation: gymno. (gymnosperm), pelta. (peltasperm), ginkg. (ginkgophyte), cycad. (cycadophyte), gigan. (gigantopterid), corda. (cordaitalean), spheno. (sphenophyte), glosso. (glossopterid), Tr. Lyco. (Triassic lycopod). This plant classification is only applicable to the late Permian to Middle Triassic and cannot be directly applied to other time intervals. The palaeogeographic reconstructions are from the PALEOMAP Project (http://www.scotese.com/Default.htm).

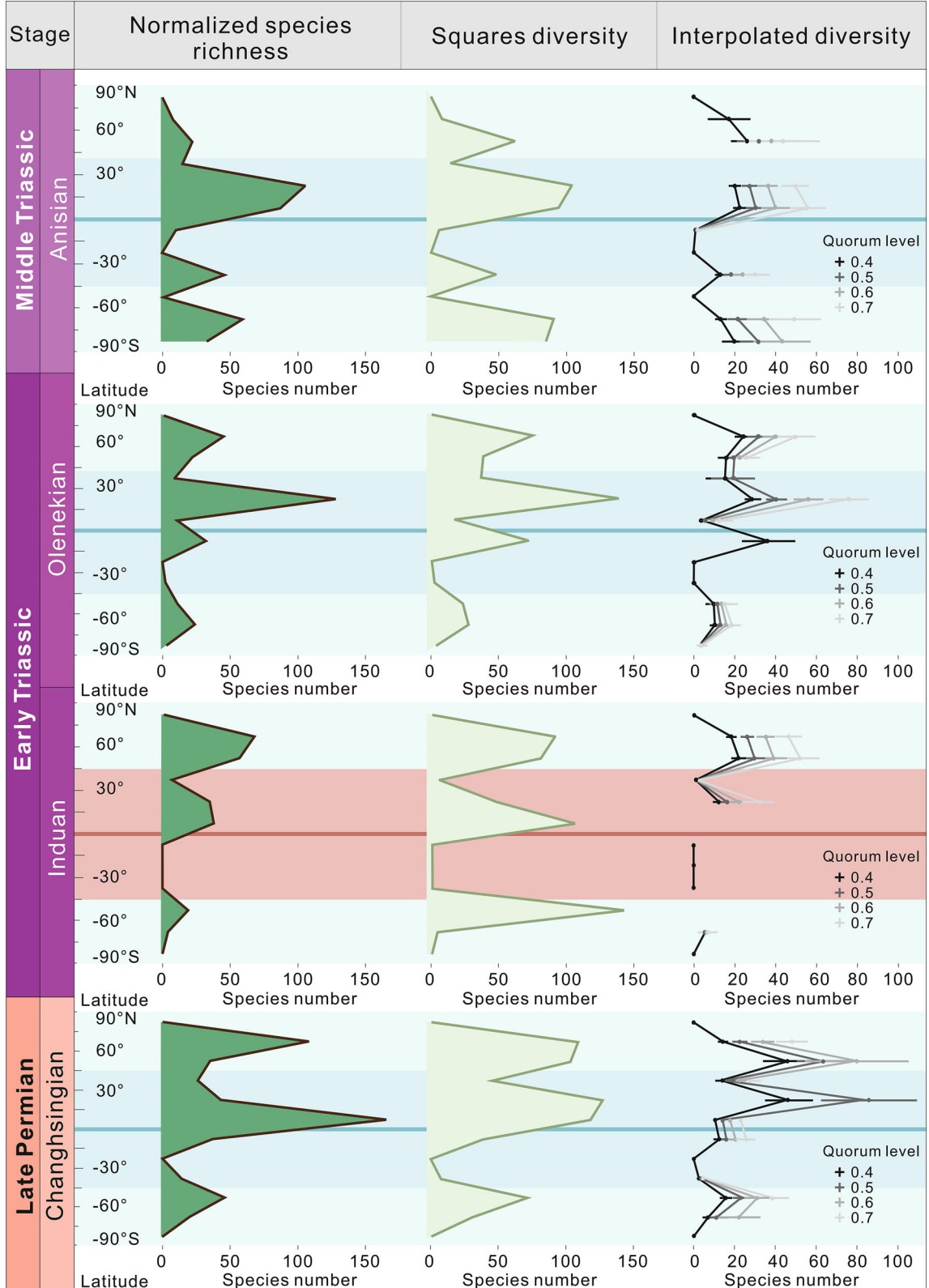

**Fig. 2 | Normalized plant macrofossil species richness, squares diversity and interpolated diversity.** Plotted in 15-degree latitude bins for each stage. Horizontal coordinates show taxa number and vertical coordinates show latitude. Shading shows 'high latitudes' (-45°–−90° and 45°–90°) and 'low-middle latitudes' (−45°–45°). Bins with less than three species have been plotted as '0', while missing points indicate an estimated diversity of more than three times the observed value. Error bars indicate 95% confidence intervals.

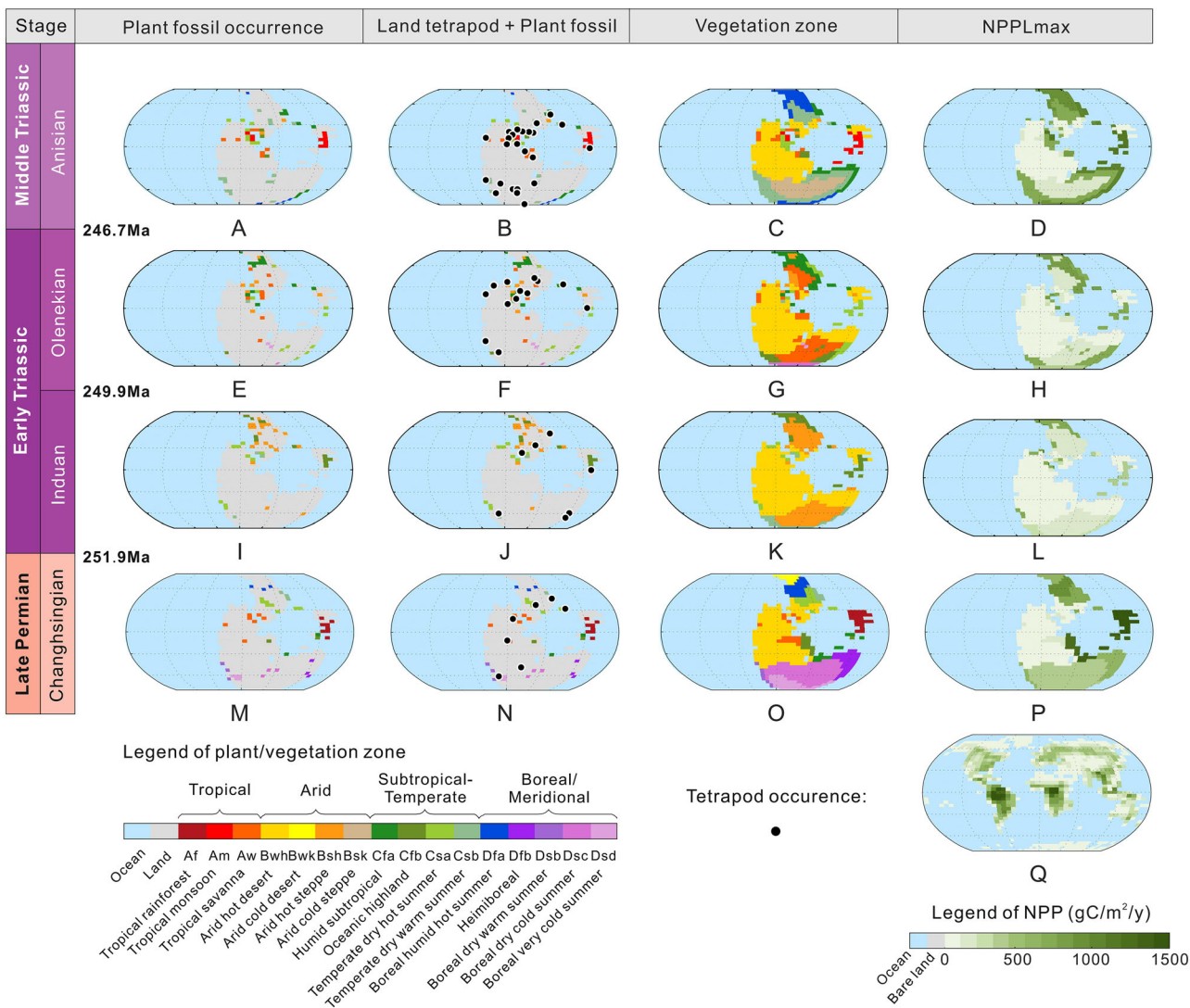

**Fig. 3 | Late Permian to Middle Triassic maps of plant and land tetrapod fossil records, vegetation reconstruction and Net Primary Productivity (NPP) distribution.** See Methods for details. 'Plant fossil occurrence' represents raw plant fossil data (Supplementary Table S1 and S2), 'Land tetrapod+Plant fossil' represents terrestrial tetrapod occurrence data superimposed on land plant fossil data (Supplementary Table S3), 'Vegetation Zone' is the interpolation of that data using lithological indicators of climate zonation (Supplementary Table S5), and NPP is reconstructed based on the present day (Supplementary Table S6 and S7). End Permian Changhsingian: **M, N, O, P**; Early Triassic Induan: **I, J, K, L**; Early Triassic Olenekian: **E, F, G, H**; Middle Triassic Anisian: **A, B, C, D**; Modern world: **Q**. All maps are centered around 0,0. Tetrapod data is from Allen et al.[36] The palaeogeographic reconstructions are from the GEOCLIM model.

coverage[20,21,42,43]. During the PTME, high latitude areas such as Siberia, and high-altitude areas at low to middle latitudes including parts of China, the Middle East and Euramerica, provided a refuge, while the expansion of high temperatures and seasonal aridity saw the loss of most lowland and marsh plants in the lower latitudes of the Early Triassic[20,25,30,35,44-46]. According to the plant functional traits recorded in macrofossils, the pre-extinction lowlands from low to high latitude around the Tethys Ocean were covered by arborescent forests with a canopy layer possibly reaching 50 m high, which were replaced by herbaceous ground covers with heights from 0.05 to 2 m in most low to middle latitude areas (Fig, 2; Table S5). In parallel to the reduced plant height and floral spatial complexity, leaf size also decreased in both compound and simple leaf groups, inferring that high productivity forests were replaced by smaller biomass communities with lower productivity in lowlands[20,30,35] (Table S5). Thus, while plant global diversity may not have suffered a catastrophe at the PTME in upland areas[45], the diversity and biomass in low-middle latitude lowland areas was substantially reduced[20,23,25]. After the inhospitable Induan stage (251.9–249.9 Ma), plants gradually migrated out from refuge areas

during the Olenekian stage (249.9–246.7 Ma). Further recovery in the Middle Triassic Anisian stage (246.7 - 241.5 Ma) saw tropical biomes reappear at low latitudes, as well as the resumption of coal deposition[20,23] (Fig. 1).

## Reconstructing plant productivity

To produce a map of palaeo-productivity from our distribution of biomes, we rely on evidence from the present[47]. Key Carboniferous plants likely had growth and transpiration rates similar to modern angiosperms[47]. Therefore, we assume that Permian to Triassic plants, either related or analogous to these Carboniferous species or resembling modern plants, functioned like today's angiosperms and gymnosperms[47-49] (see "Methods" 8). The Net Primary Productivity on Land (NPPL) of each grid cell in our palaeogeographic reconstructions (Fig. 3) was determined using these nearest living plant functional type that shares a similar plant size and form, basic spatial structure, function, diversity, geographic location and climate zone. Within each plant functional type, there is normally more than one recent flora that fits the requirement of each palaeo flora, and these recent floras are

arranged from high to low NPPL to run the sensitivity tests, with only the highest and lowest members shown in Table S7. Here, we aim to generate the general land vegetation productivity trend across the PTME in a consistent comparative system within the palaeo- and modern plant functional types rather than using vegetation modeling. This is because although dynamic vegetation models have simulated similar NPPL to our Changhsingian estimates, they do not yet incorporate experimental data on plant response to extreme hothouse conditions like those of the Early Triassic[49,50].

Our NPPL estimates based on these palaeogeographic reconstructions show fluctuations from ~54.4–62.5 Pg C/yr in the latest Permian Changhsingian, to a low of ~13.0–19.7 Pg C/yr in the Early Triassic Induan (a loss of ~70%), followed by Olenekian values of ~25.0–32.2 Pg C/yr, with ~53.8–63.5 Pg C/yr in the Anisian. Before the PTME, the global terrestrial productivity gradient correlated with latitude, with the highest values in the tropics, similar to the modern world[48,51]. However, this trend dramatically reverses after the PTME, as regions of high productivity migrated from low-to-high latitudes, before gradually re-establishing the previous gradient during the Olenekian and Anisian stages (Fig. 3). Comparison of fossil-based reconstructions are broadly consistent with simplified plant thermal adaption modeling[22,48] and show loss of low to -mid latitude forests, survival at higher latitudes, and a major productivity collapse post-extinction.

### Modeling plant effects on long-term climate

Our fossil-based reconstructions thus far represent 'biogeographic productivity', which we define as a productivity metric that does not consider the effects of $CO_2$ fertilization. This mechanism would be expected to increase productivity, given generally higher $CO_2$ in Earth's past[19,51,52] and especially in the Early Triassic[7]. To test the biogeochemical and climatic effects of these shifts in plant biogeography while also taking $CO_2$ fertilization into account, we use our palaeobiogeographic reconstructions as inputs to the *SCION* Earth Evolution Model[52,53]. *SCION* is a global climate-biogeochemical model that links steady-state 3D climate[54] and surface processes to a biogeochemical box model[55]. It calculates continental weathering rates at each grid point on the land surface based on local temperature, runoff and erosion rates, as well as an assumed biotic enhancement factor (*fbiota*). In order to calculate Net Primary Productivity on Land (NPPL) in *SCION*, we used the biogeographic productivity estimates from our maps (Fig. 3), and then added an established function for the $CO_2$ fertilization effect[56] (see "Methods" 10), based on the modeled $CO_2$ concentration at the current model timestep. The land vegetation productivity after the $CO_2$ fertilization effect has been applied is named NPPL$_f$.

We then modified the biotic weathering enhancement factor (*fbiota*) based on the fossil-based NPPL$_f$ in each grid cell, allowing for a 4-fold enhancement between the most and least productive grid cells as a conservative estimate (see refs. 57–59 for a range of estimates of this factor, and Methods 10 and 11 for model runs with different factors). In order to modify the global rate of organic carbon burial, we summed the total fossil based NPPL$_f$ for each time period and used this to scale the flux of terrestrially derived organic carbon burial (see Methods 10 for details). Marine productivity and organic carbon burial is also calculated in the model based on limiting nutrient availability[52]. Aside from these biotic changes, the *SCION* model retains the Phanerozoic scale forcing information from previous standard runs[52], including background tectonic $CO_2$ degassing. The only abiotic alteration to the model was to include additional $CO_2$ degassing from the Siberian Traps[60], which accurately reproduces the shorter term (~500 kyr) carbon isotope perturbations across the PTME (Fig. 4D). The model is initialized at 300 Ma with present day $CO_2$ concentration, but quickly achieves a long-term steady state equivalent to the Phanerozoic-scale model[52].

## Results

Figure 4 shows the *SCION* model results through the latest Permian, and the Early and Middle Triassic, both with and without the inclusion of our palaeo-vegetation constraints. In the control run (dashed black line), the biogeographic NPP of each continental grid cell is kept constant at 420 g C/m$^2$/yr to produce an overall productivity similar to our late Permian fossil-constrained vegetation map, and all changes in the model environment are driven by abiotic forcings, such as background tectonic degassing rates and Siberian Traps degassing. The major features of this default run are the spikes in $CO_2$ concentration and temperature (Figs. 4E, 4G), and the accompanying $\delta^{13}C$ excursion (Fig. 4D), driven by Siberian Traps degassing. The magnitude of the isotope excursion is consistent with the geological record and previous modeling[5,60], and $CO_2$ concentration rises from about 1,500 to 3,000 ppm, with a corresponding increase in equatorial surface temperature of about 2 °C. The high background $CO_2$ level and relatively small temperature increase are both features of the low climate sensitivity in the *FOAM* climate model[61], which provides the steady state 3D climate emulator for *SCION*. Thus, we expect that a more complex model might allow for a more dramatic temperature increase and lower overall $CO_2$ levels, as suggested by some proxy data[7,10]. However, no amount of climate model complexity can account for the data-model mismatch during the Early Triassic, where model temperatures decline immediately after the cessation of Siberian Traps emissions. Because *SCION* has a single-box ocean, it does not balance sub-million-year alkalinity and shallow sea carbonate deposition as accurately as multi-box models, in which $CO_2$ levels decline even more rapidly[60].

When included in the model, the loss of vegetation productivity from the end Permian through the Early Triassic, and the related effects on continental weathering, result in a sustained high atmospheric $CO_2$ content[7] and high Early Triassic temperatures[10] (green line in Fig. 4C). In these model runs, the reduction in terrestrial organic carbon burial and nullification of silicate weathering result in $CO_2$ levels stabilizing at around 7000 ppm, with maximum equatorial surface temperatures of up to 33–34 °C over a ~5 Myr period, which is consistent with proxy inferences (Fig. 4)[7,10]. As before, low climate sensitivity in our model results in predictions of high $CO_2$ concentrations, although the magnitude of predicted increase (~4 fold) is broadly equivalent to that suggested by proxies[7]. The modeled Early Triassic $\delta^{13}C$ level (green line in Fig. 4D) is also around 2–3‰ lower than the control run (black dashed line in Fig. 4D), generally improving the fit to the geological record[10] (blue solid line with dots in Fig. 4D). Two exceptions to the data-model fit are the Induan–Olenekian (Dienerian–Smithian) boundary and the late Olenekian (early Spathian), which are marked by transient positive carbon isotope excursions that may have been driven by increasing marine productivity, transgression, or marine anoxia[8,10], none of which are considered in our stage level study. Model strontium isotope ratios, which are influenced by continental weathering fluxes and source lithologies, are not greatly affected by the inclusion of vegetation collapse, but show a slightly greater rise between the Changhsingian and Anisian as weathering migrates to higher latitudes and away from low latitude suture zones with low $^{87}Sr/^{86}Sr$ values[62] (Fig. 4C, see Supplementary Fig. S4 for lithological map, and Methods 11). Our *SCION* model demonstrates how the prolonged hothouse environment could have been terminated by progressive terrestrial ecosystem recovery, starting in the Olenekian but accelerating in the Anisian, which is also consistent with the observed uptick in $\delta^{13}C$ values across the Olenekian–Anisian boundary and the

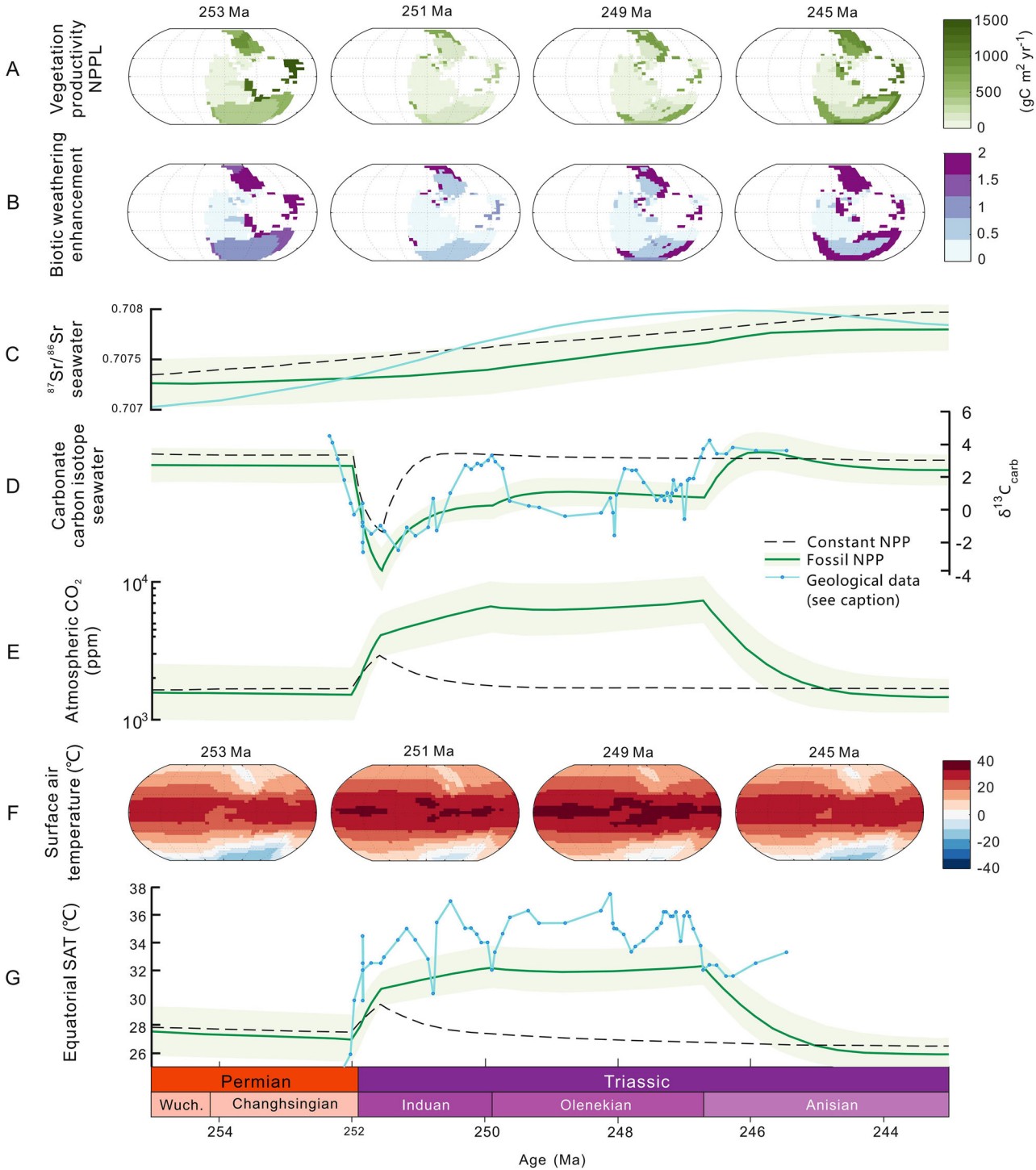

**Fig. 4 | Climate-biogeochemical model driven by terrestrial Net Primary Productivity (NPP) changes.** The vegetation NPP is prescribed onto the land surface in the SCION model (**A**) and affects the model calculations for organic carbon burial and the biotic enhancement of continental weathering (**B**). The model is run with (green solid line) and without (black dashed line) the fossil-prescribed NPP, where both models include the Siberian Traps degassing. **C** Ocean $^{87}Sr/^{86}Sr$ compared to McArthur et al.[116] (blue solid line). **D** Carbonate $\delta^{13}C$ compared to the dataset of Sun et al.[10] (blue solid line with points). **E** Atmospheric $CO_2$. **F** Surface air temperature at chosen timepoints. **G** Equatorial surface air temperature (SAT) compared to the equatorial SSTs of Sun et al.[10] (blue solid line with points). All the geological records have been correlated based on chronostratigraphic correlation with the GTS (2020). The palaeogeographic reconstructions are from the GEOCLIM model.

cooling which occurred during this time[10]. This dynamic fits with broader evidence for a more benign environment during the re-establishment of diverse ecosystems in Middle Triassic terrestrial settings[20,35]. A 'CO$_2$ fertilization' effect occurs in the model in the early stages of warming, before atmospheric $CO_2$ and land surface temperatures reach saturation point for C3 plant photosynthesis.

However, this increased local productivity in some areas was insufficient to offset the global decline in biomass abundance.

## Discussions
In our model scenario, the reduction in continental silicate weathering intensity caused by decreased plant productivity had a greater impact

on increasing atmospheric $CO_2$ than the direct effect of a decline in organic carbon burial (see Fig. S6). This is because while the large reduction in terrestrially derived organic carbon burial acts to increase $CO_2$ levels, it also decreases atmospheric oxygen levels and redistributes nutrients to the ocean, meaning that more marine organic carbon is produced and preserved, and less fossil organic carbon is weathered. Several limitations of our approach may be responsible for under-predictions of the magnitude of temperature rise. The negative feedback on the organic carbon cycle may be too strong, which may be why SCION fails to replicate more rapid variation in Phanerozoic atmospheric $O_2$[52]. Additionally, the weathering of sedimentary organic carbon likely increases with temperature[63], which is not accounted for in the model, and may nullify these negative feedbacks further. A further uncertainty in our modeling is the degree to which plants amplify continental weathering, as shown in Figure S6, with the 'best guess' values from Phanerozoic-scale models of plant weathering strength (being a 4–7 fold enhancement[52,57].) producing different magnitudes of warming. Previously suggested mechanisms for Early Triassic warmth, such as limited erosion rates or amplified reverse weathering, also potentially played a part in the extreme warmth[11,14]. They are not included in our model due to the difficulty in quantifying their magnitudes and their timeframes of operation, but they could feasibly raise $CO_2$ and surface temperature further.

Our study provides a quantitative estimation of changes to global palaeo-plant biomass and corresponding long-term environmental impacts. Through our modeling, we show that the large and prolonged decrease in tropical plant productivity in the Early Triassic likely resulted in a world that was lethally hot by Phanerozoic standards, a consequence of substantially weakened terrestrial carbon sequestration rates. These conditions persisted for nearly five million years and cooling was only achieved as plant productivity began to increase in the Middle Triassic. We believe this case study indicates that beyond a certain global temperature, vegetation die-back will occur, and can result in further warming through removal of vegetation carbon sinks. Our study demonstrates that thresholds exist in the Earth system that can accelerate climate change and have the potential to maintain adverse climate states for millions of years, with dramatic implications for global ecosystem behavior.

## Methods

### Age dating of plant records

The geological timings used in this paper are from the Geological Time Scale (GTS) 2020. Selected study areas (and sites) are the Kuznetsk Basin in Siberia, Junggar Basin (Dalongkou section) in Xinjiang, NW China, eastern Yunnan and western Guizhou in SW China, Utah in USA, Germanic Basin in western Europe, small Tethyan continents, now incorporated in southeastern Asia, Turkey in the Dead Sea area, Kashmir in NW India, Karoo Basin in South Africa, Sydney Basin in Australia, Prince Charles Mountains in Antarctica, and Argentina, covering the published plant fossil bearing areas from various latitudes (Fig. S2). We reviewed the published chronostratigraphic correlations between the floral records, other environmental events, and the lithological Permian-Triassic Boundary in each area to determine the global pattern of plant evolution. Chronostratigraphy of the non-marine strata is correlated by fossil assemblages including animals and plants, detrital zircon ages, and geological events recorded by geochemical proxies. The detailed records and analysis of each location are provided in the Supplementary Information text 1.

From the evidence noted in the SI part 1, it is clear that end-Permian terrestrial crisis happened in the late Changhsingian over an interval starting 750 kyrs before the PTB up to shortly after the boundary[20,64–67]. In high latitudes, the macrofossil records show *Cordaites* in Siberia and the *Glossopteris* flora in Gondwana disappeared in the mid to late Changhsingian[68,69]. An abundant flora of ferns, seed fern peltasperms, cycadophytes and conifers survived through the PTB in

the high latitudes[20,68,69]. In our analysis, plant fossil occurrences were noted at the stage level giving the impression that all the plant changes were near the PTB.

### Plant macrofossil and palynology data normalization steps

Plant macrofossils are typically fragmented into different parts (organs) prior to fossilization, with each part often named separately using Linnean binomials[29,70]. We normalized the dataset to correct for duplications in which different parts of the same plant are included under different species or genus names, and ensured the same taxa with different morphological names could be linked. In normalization, organs such as species or genera of seeds, trunks, roots and leaves are removed from the dataset if another organ from that plant group is more likely to reflect the whole plant taxonomy, so that each whole plant is counted only once[29,70]. An example is the diverse trunk group of tree lycopods, where species of *Lepidodendron* are used as they are abundant and systematically informative[29,70], rather than other organs produced by the same plant, including cones, sporophylls (fertile leaves) or roots (see ref. 20 for detail). For diverse leaf groups, for example, ferns and sphenophytes, leaf species or genera are used, as these fossils typically lack more distinctive organs with suitable preservation. Indeterminate species denoted as "sp." of an existing genus are regarded as likely to be poorly preserved examples of the existing species of that genus, and are deleted. If the indeterminate species denoted as "sp." is the only species in that genus, they are counted as a single, unnamed species. Normalized plant macrofossil species data is listed in Table S4. In addition to these, palynological occurrences are also considered. Most of the palynological data are linked with plant macrofossils at family level, with a few spore and pollen taxa preserved in-situ within their parent plant for which the genus and species names of the parent plants are used.

### Plant macrofossil species extinction magnitude

All the species occurrences presented are based on the normalized data (Table S1). Longitude and latitude for each fossil location are listed in Table S2 and S3. The high latitude area is defined to be >45 degrees north and south of the equator, while low-middle latitude area is <45 degrees north or south. This definition is for this study only and is not climate specific. The range of plant fossils in each stage was checked and extended for calculating the extinction magnitude over a global high-latitude and low−middle latitude area. The extinction magnitude for each stage is the extinct species number compared to a later stage, minus the total normalized species number of this stage[20]. See the extinction magnitude results in Table S4. The extinction magnitude of the Anisian is not estimated.

### Plant latitudinal diversity calculation and influence of sampling density

To investigate the influence of plant fossil sampling completeness on our estimates of diversity, squares and interpolation methods were applied to our normalized plant macrofossil occurrence data. As for the raw data, squares and interpolation were applied to 15° latitude bins for the Late Permian (Changhsingian) to Middle Triassic (Anisian). Coverage-based interpolation uses the abundance structure present within samples, to either subsample or extrapolate diversity estimates to particular levels of sampling completeness, known as quorum levels[36,71,72]. This was applied using the R package iNEXT[36]. Squares is an extrapolator based on the proportion of singletons in a sample and is thought to be more robust to biases associated with small sample sizes and uneven abundance distributions[73,74].

Throughout the interval, the raw, squares and interpolated diversity estimates generally show similar latitudinal patterns, suggesting that sampling is not a strong influence on our inferred latitudinal diversity gradients (Fig. 2). However, many of the points in the interpolated curves were removed due to over-extrapolation, which

indicates that many of the spatio-temporal bins may be under-sampled. Our results indicate that during the Induan, the highest plant diversity was found in the high latitudes, particularly in the northern hemisphere. However, during the Changhsingian, Olenekian and Anisian, we see higher diversity levels at tropical latitudes, suggesting that the latitudinal diversity gradient had reverted to a situation similar to that of the present day one to two million years after the PTME.

## Plant functional trait evaluations

Plant functional traits are stable morphological, anatomical, and compositional characters that have evolved under specific climates and environments, linking plant physiological processes to the Earth's biogeochemistry and physical evolution[31,32,75,76]. The plant functional traits of late Permian to Middle Triassic fossil plants are not well studied. Here, we aimed to determine the habitat of the fossil plants, the climatic zone in which they lived and to semi-quantify the biomass of the flora. We selected plant traits including plant growth form, reconstructed plant height, which indicates the spatial structure of the flora, and leaf size, which indicates the potential biomass of the flora. For water, carbon and nutrient cycling in the plant, leaf shape, vein pattern and density, and cuticle thickness are considered, which determines the plant's moisture preference, drought resistance, and productivity[31,32]. Cuticle thickness of present Ginkgo is positively related to productivity, although this is not further explored in our dataset due to the lack of experiments on these Permian-Triassic plant's recent analogs[31].

Leaf size and vein density were measured using ImageJ, and only the largest and most complete leaves of each taxon are listed in the table S5, with fossil plant data collected from references in the supplementary table references. Other traits, including plant form, whole plant height, vein type, and cuticle thickness, are semi-quantified and are based partly on the reconstructed fossil plant, including *Lepidodendron*, Triassic herbaceous lycopods, sphenophytes including *Calamites*, gigantopterids, glossopterids, and ginkgophytes. Features of plants from which reconstructions are unknown are inferred from reconstructed relatives in the same genus or family; these plants should be further investigated in the future to characterize them more accurately.

Fossils with measurable leaf size are listed in Table S5 which covers low to high latitudes. Interpretation of the floral climate zone and vegetation landscape information are based on all the micro and macro plant fossils, including those without measurable leaf size. The general concept is that plants with higher height, larger leaf size, higher vein density, and more complicated vein system are of higher biomass and relied on greater humidity for transpiration. Floras with a higher proportion of these plants normally have higher species diversity and spatial structure complexity, which suggests higher productivity. Details of the plant trait relationships are outlined by ref. 31.

## Flora characterization by clustering and morphological group

To analyse the character of floras from the late Permian (Changhsingian) to the Middle Triassic (Anisian) for comparison and matching between floras with certain functional analysis and missing information, family level clustering was used to group the floras with the normalized plant fossil data. The clustering result is based on the Euclidean method. The plant systematic information comes from the listed literature, with additions from the Global Biodiversity Information Facility (GBIF) https://www.gbif.org/ database which were checked against the literature to ensure their accuracy. The taxonomic affinity of most spore and pollen taxa are unknown, and so only plant macrofossil data was used in clustering, and the palynology data was only used in the morphological group diversity analysis. To show the uncertainty of the clustering results, we list the plant species number

after each flora in Fig. 1B. Unsurprisingly, the clustering results for flora with fewer taxa were less reliable and more crowded together. As an auxiliary method to clustering, we counted the plant species number in each morphological group (see the fourteen morphological group classifications below), then calculated the proportion of the species number in each morphological group within floras to directly show the character and to construct a representative pie chart for each flora. For floras with fewer taxa, we adjusted the location of each flora in the clustering tree manually, according to the character shown by the morphological group diversity.

Plants were divided into six habitats and fourteen groups, including four arid upland types: conifer, gymnosperm (for seed plants where systematic class/group is uncertain), peltasperm and seed fern; three humid upland types: cordaitalean, ginkgophyte, cycadophyte; one rainforest type: gigantopterid; two humid types: fern and 'fern2' (for taxa that could be either ferns or seed fern); two marsh types: sphenophyte and lycopod; one cold type: glossopterid – normally reported in boreal Gondwana; and one arid lowland type: herbaceous lycopod. This classification is only for the plants included in this study and must be carefully applied to other time intervals by checking the habitat of the fossil plants in detail. Flora dominated by one habitat group was classified into the corresponding climate zone, and flora with more than one habitat group was defined as a mixture. In this step, we also took palynology data into account. The group information of the in-situ spore and pollen producing plant were counted[77]. For flora with both plant macrofossil and palynology data, we chose the dataset which contains more information. In Fig. 1B, flora with more than 150 taxa, such as the Changhsingian South China flora, have the biggest pie chart area, while flora with less than 10 taxa, like the German flora, have the smallest pie chart. After the plant function traits, habitat, and clustering analysis, the character of the flora from the End Permian to the Middle Triassic was systematically studied and classified into climate zones as shown in Table S5 and S6.

## Palaeogeographic reconstruction

To reconstruct the spatial vegetation map, we assembled a database of fossil locations, plant macrofossil, palynology, and terrestrial tetrapod data for our time periods (Table S1–S3). To account for plant refuges or Mesozoic gymnosperm cradles that may not be represented in the fossil database, we first extended the plant megafossil range in each basin. For example, voltziales and peltasperms were found in North China and Euramerica before the PTME and reappeared in the Middle Triassic but were absent in Early Triassic strata, so we extended the ranges of those surviving gymnosperms[78–80]. Plants that were dominant after the Early Triassic but had already appeared in end Permian strata in Argentina, India and Northeast China have all had their ranges extended through the Early Triassic[69,81–83]. Secondly, the palynological data tends to better record information on upland floras, while the plant megafossil data records primarily lowland taxa[20,30]. We detected hidden upland floras in South China, China Xinjiang, South Africa, Antarctica, and Australia based on gymnosperm pollen evidence after the PTME[21,84–91]. Our analysis included all the plant data from macro and micro floral records in all sedimentary facies, to avoid using the local information and to represent information from the whole basin. Thirdly, terrestrial tetrapod data was used to infer the occurrence of plants in regions without a plant fossil record[24,36]. Generally, terrestrial tetrapod occurrences in our study coincided with occurrences in the plant fossils, except for the Olenekian record in America and Canada. Therefore, vegetation type in those areas at this time was partly inferred from the tetrapod information alone on the presumption that plant primary producers were necessary in these regions to support vertebrate communities.

The fossil locations were then reconstructed to their time of deposition using *GPlates*[92]. Because the older palaeogeographic reconstruction used in *SCION*[93] has no available set of rotation files, we

used the reconstruction files of ref. 62, whose reconstruction at ~250 Ma is very similar to that of ref. 93 This allowed us to place fossil locations in an internally correct position at 250 Ma. However, minor manual manipulation was needed to then map some of these locations to their correct corresponding positions in the *SCION* land-sea maps.

## Vegetation productivity reconstruction

Global vegetation was reconstructed by extension of fossil flora data across appropriate climate zones indicated by a sedimentary climate map[94]. In arid areas, plant fossil extrapolation is not applied[95]. Extrapolation was not carried out at the boundaries of humid and arid environments in places that lacked supporting mineralogical data. For example, the Early and Middle Triassic low-latitude inner Pangea continent is inferred to have been arid savanna or steppe, based on the available fossil record and lithological climatic indicators. Fossils from more productive biomes which are found nearer the coast are restricted to this setting and not extended far into the continental interior where climate is arid (Fig. 3).

Three principles are used for functional comparison between ancient and recent floras to estimate palaeo-productivity: firstly, recent floras must have a similar structure to the ancient flora we wish to imitate, so we compare the reconstructed size of the fossil plants including height and leaf size of the individual plant and the spatial structure of the flora with the recent analog. For example, the end-Permian tropical South China floral canopy is dominated by tree lycopods *Lepidodendron* and sphenophytes, fern and gigantopterid understory, showing the highest complexity in forest spatial structure among all the floras from the late Permian to Middle Triassic. The dominant gigantopterids have giant leaves with typical rainforest drip-tip structure, complicated vein systems and high vein density, and thus recent rainforest is chosen as an analog for the late Permian South China area, Southeast Asia, China Xizang, and Turkey which shared high similarity in taxa composition[33]. Secondly, the recent and ancient floras should be in the same climate zone. For example, the latest Permian South China tropical forest was a large, low-latitude island, and so present-day, large tropical islands like Indonesia and Thailand were chosen over (for example) continental Brazil. Each palaeo flora has suitable recent floras sharing similarity in plant function, climatic and geographic zone. Thirdly, the chosen flora should fit in the global diversity and NPP gradient at a similar place to the ancient flora. For ancient floras with clear functional trait records, we compare the NPP between floras from the late Permian to Middle Triassic by the traits mentioned above. We also use present-day data to confirm the hypothesis that plant diversity has a positive correlation with productivity, which generally fits with our normalized fossil results[96,97]. For instance, the late Permian (Changhsingian) tropical South China flora is matched with a present day high-diversity and highest-productivity biome in the chosen recent non-continental tropical rainforest functional group range, that of present-day Thailand. The late Permian to Middle Triassic fossil plants' inferred habitat, climate zone and landscape are shown in Table S5. The calculated NPP of each ancient flora is listed in Table S6 and details of corresponding recent flora are in Table S7. The reconstructed NPPL is marked as NPPLfossil.

Although this study makes progress in comparing the physiological and functional difference between palaeo- and recent plants, more detailed studies are still required. For example, the influence of the interaction between plants and other organisms including mycorrhiza and insects are not considered, neither are soil texture differences between the Permian-Triassic and present day. Although our results have associated uncertainties, the very large change in biomass over the PTME is very likely much larger than these potential errors. Nevertheless, without detailed study on palaeo plant physiology in other deep time periods, the methods mentioned above should not be directly applied to other timeframes.

## Taphonomic influence on fossil preservation

In our initial steps to reconstruct land biomes and productivity, we did not extend the distribution of land plants to regions lacking climatic or mineral records of hospitable environments. For instance, the central areas of the Pangea supercontinent, characterized by evaporites rather than coal deposits, were identified as deserts or barren lands where plant growth was presumed unlikely. However, while the absence of fossil records in some areas may result from local biome extinctions, it could also be due to poor preservation conditions caused by insufficient water availability[98]. To assess the influence of taphonomic bias on our reconstructions—essentially, whether areas without fossil records truly lacked local biomes or merely lacked preservation conditions—we employed two different approaches.

First of all, we assessed the sedimentary facies and strata thickness of basins from various latitudes and locations[99] (Fig. S3). For example, in low latitude South China, the fossil-plant-absence zone occurs in the Kayitou Formation, formed in a shallow lake or floodplain environment[100]. The central European basin recorded a warm seasonal humid climate in the Early Triassic, which is likely a response to global warming after the PTME in inner Pangea[101]. In high latitude South Africa and Australia, sedimentation patterns and occurrences of green algae indicate widespread ponding environments through the plant 'dead zone'[85,102–104]. While these are strictly local examples, they indicate the presence of waterlogged, swampy environments suitable for plant preservation in the Early Triassic, suggesting that taphonomy is not the primary cause of the absence of fossils, and that this may reflect a genuine reduced abundance of plants after the PTME in these areas[20]. Additionally, frequent wildfires point to intensified seasonality or seasonal aridity, likely reducing lowland plant habitats and contributing to the sparse record of "hidden" upland plants[2,20,21,67,85].

Second, we explored the potential existence of hidden "refuges" or "cradles" using our climate model, assuming plants could survive in grid cells with land surface temperatures below 40 °C and runoff above 0 mm/yr, similar to the conditions required by most modern plants[105,106]. The results suggest that suitable environments for plant survival existed in high-altitude and coastal areas, even in some low-latitude regions, such as South China, North China, Xinjiang, Europe, and Central Asia. These findings are consistent with our reconstructions based on fossil records. For example, the South China Induan plant macrofossil record is dominated by the herbaceous lycopod *Tomiotrobus* with a maximum height of 0.2 meters, while coeval palynological data suggests a hidden upland gymnosperm flora[20,21,107]. Therefore, the Induan flora in South China is compared with recent Australian shrubs or the seasonal dry subtropical forest in China Yunnan Province which is herbaceous and shrub-dominated with sparse tree cover.

## Climate-biogeochemical modeling

To investigate the effects of vegetation change on Early Triassic climate, we ran the *SCION* Earth Evolution Model[52]. We removed the equation which calculates terrestrial vegetation biomass (as a single global number) and replaced this with values based on our reconstruction, mapped onto the model continental surface. To calculate NPPL$_f$ we multiplied the reconstructed NPPL$_{biogeographic}$ by a factor representing $CO_2$ fertilization. This follows the Michaelis-Menton formulation used in the *GEOCARB* biogeochemical models[56,108]:

$$Fert = \left( \frac{2 \cdot RCO_2}{(1 + RCO_2)} \right)^{0.4} \tag{1}$$

where $RCO_2$ is the atmospheric $CO_2$ concentration relative to the preindustrial 280 ppm.

Plants play a crucial role in terrestrial weathering[109]. Present-day global-scale observations show that silicate weathering is linearly related to plant NPPL$_f$ (Fig. S5)[48,110,111]. We altered the model parameter $f_{biota}$, which represents the biotic enhancement of continental weathering (again a single global average in *SCION*), to make this dependent on the local vegetation in the following way:

$$f_{biota} = 0.002 \cdot NPPL_f + 0.25 \qquad (2)$$

Functionally, this returns a value tending towards 0.25 when NPPL$_f$ is very low, and a linear scaling with NPPL$_f$ when NPPL$_f$ rises. The choice of 0.25 relates to the four-fold enhancement between simple ground covers and higher plants used in first-generation long-term carbon cycle models like *GEOCARB*[56,108], and based on field and laboratory studies[53]. We vary this 'preplant' factor between 0.15 – 1 and modify the linear scaling to run sensitivity tests for various plant weathering abilities (Fig. S6). In all formulations, the scaling factor for NPP is chosen to return present day global weathering rates for the model present day integration.

### Strontium isotope $^{87}Sr/^{86}Sr$ lithology

The strontium isotope composition of river water has a strong local lithological control, so to simulate the Sr isotope record we imposed basic lithological classes on the model continental surface using the locations of continental arcs, LIPs and suture zones from the literature[62,63,112] (Fig. S4). These zones were then prescribed representative Sr isotopic values[52,57,113–115].

## Data availability

The normalized plant and land tetrapod data taxa list and occurrence are provided in Supplementary Table S1–S7. The normalization details are available from Zhen Xu on request.

## Code availability

The *SCION* model is freely available at https://github.com/bjwmills/SCION and the modified version used for this work is archived at https://github.com/ZhenXuJane/SCION_Xu2025.

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

## Acknowledgements

We thank Q. Xue, W.J. Ran, H. Li, S. Xiao, W.C. Shu, Y.H. Guo, N. Peng, B.B. Li, W.J. Lin, M. Fan, X.J. Wang, M.J. Zhang, Y.Y. Tian for assistance in the field work. We thank those who have contributed to the plant and tetrapod data obtained from the Palaeobiology Database and the Global Biodiversity Information Facility. We thank Lee Kump and Isabel Montãnez for their contribution in suggestions and discussions of this work. This work is funded by the Natural Science Foundation of China (grant 92055201, 42430209), H.F. Yin, J.X. Yu, Z. Xu, P. Wignall, A. Dunhill and J. Hilton are funded by the UK Natural Environment Research Council Eco-PT Project (NE/P0137724/1), which is part of the Biosphere Evolution, Transitions and Resilience (BETR) Program, J. Shen is funded by the Natural Science Foundation of China (grant 92479203), A.S. Merdith is supported by ARC DECRA Fellowship DE230101642, Z. Xu is funded by the China Scholarship Council (202106410082), Z. Xu, K. Gurung, and B.J.W. Mills are funded by the UKRI project EP/Y008790/1, B.J.W. Mills and S.W. Poulton are funded by the UK Natural Environment Research Council (NE/S009663/1), B.J. Allen is funded by the ETH+ grant (BECCY). A. Dunhill and P. Wignall are funded by UKRI Natural Environment Research Council Grant NE/X012859/1 and A. Dunhill is funded by UKRI Natural Environment Research Council/National Science Foundation Grant NE/X015025/1.

## Author contributions

Z.X., J.X.Y., H.F.Y., and B.J.W.M. designed the study. Z.X. collected the plant dataset, and Z.X. and J.H. normalized and analyzed the plant dataset for the vegetation reconstruction. B.J.A. calculated the plant Squares and interpolated diversity. A.S.M. produced the python code for the palaeogeographic reconstruction and the spatial lithology map for the strontium isotopes. B.J.W.M. and Z.X. modified and ran the *SCION* model. Y.G. and Y.D. provided *FOAM* climate model datasets and discussion of weathering processes. Z.X. and B.J.W.M. wrote the paper with contributions from J.H., P.B.W., S.W.P., A.S.M., A.M.D., B.J.A, J.X.Y., H.F.Y., K.G., J.S., D.S., Y.G., Y.D., Y.X.W., and Y.G.Z.

## Competing interests

We declare that none of the authors have competing interests as defined by Nature Portfolio.
