## [Peer Review file · Nature Communications]

Early Triassic super-greenhouse climate driven by vegetation collapse

Corresponding Author: Dr Zhen Xu

Version 0:

Reviewer comments:

Reviewer #1

(Remarks to the Author)

Early Triassic super-greenhouse climate driven by vegetation collapse
Zhen Xu et al

Xu and colleagues shed light on the unusually long persistence of “super” greenhouse conditions following the Permian-Triassic mass extinction. Previous hypotheses for why it appears CO₂ could not be efficiently removed from the surface system include the reduced sediment flux of eroded material, which would limit silicate weathering, or contrastingly more rapid weathering regimes, both on the continents and in the oceans (reverse weathering). Neither satisfactorily explain why such conditions necessarily persist for 5 million years (approximately) after the event, and eventually recover during the middle Triassic. The authors counter this by proposing a model wherein the dramatic and prolonged diminishment of low-latitude plants takes centre stage. In the absence of tropical, peat-forming ecosystems substantially less CO₂ was likely to be sequestered from the end Permian. In this absence, the authors propose that Earth’s climate-carbon cycle stabilized at a much higher average temperature than the perceived norm in the run up to the extinction. In this model, it would not be until these plant biomes sufficiently recovered that sustained cooling got momentum. I generally like this idea, and think it is very worthwhile adding it to the mix of possibilities. By putting this modelling-led work out into the public domain, various geochemical and petrological proxies might later be explored across a number of expanded sections, ultimately testing whether or not this new hypothesis is genuine. I see this as an important step in this scientific process. Subject to minor line-by-line comments, I recommend publication in Nature Communications

William McMahon

Line by line comments

Should Andrew Merdith’s affiliation be updated?

Line 63. I’m weary of such compilations of sedimentation rates honestly. How can one infer how much sedimentary stasis is locked up in individual bedding planes, and how this varies throughout an expanded section? Not as easy as just dividing thickness by time.

Line 94. A few tens of thousands of years.

Line 128. How does one link tetrapod occurrences to local climate? I’m not even entirely certain whether most paleobotany-climate links are so robust. Lithological indicators seem to be the key thing here (especially the three listed, in the absence of detailed accounts of sedimentary structures in other siliciclastic lithologies which may provide some clues). If we’re talking tetrapod trackway’s here, biases are more likely to pick out localized areas of greater accommodation/sediment aggradation rates, wherein the chances of reworking substrates where the trackways form decrease. So unless you have a huge amount of data to draw from, you’d maybe be picking out these preservational hot spots (or lack thereof), rather than anything meaningful for global climate. Also, is local climate really what’s meant here. Or regional? Whilst I suspect the paleogeographic grid map is advanced, it’s surely compartmentalized on a “regional” rather than “local” level?

Line 134. Reference in incorrect place.

Line 134. “The ‘coal gap’ after the PTME was not associated with a significant loss of river or delta sediments.” Make it very clear how this is known. If it’s not from a global database of sedimentary rock volume of a particular age and lithotype (see

below for examples) I'd advise removing.

- Peters, S.E., Husson, J.M. and Czaplewski, J., 2018. Macrostrat: a platform for geological data integration and deep-time Earth crust research. *Geochemistry, Geophysics, Geosystems*, 19(4), pp.1393-1409
- McMahon, W.J. and Davies, N.S., 2018. Evolution of alluvial mudrock forced by early land plants. *Science*, 359(6379), pp.1022-1024.
- Dunhill, A.M., Hannisdal, B. and Benton, M.J., 2014. Disentangling rock record bias and common-cause from redundancy in the British fossil record. *Nature communications*, 5(1), p.4818.

I'd also suggest checking the references cited at this portion of the text (and maybe throughout) as a few seem out of place to me.

Line 146. "The extinction zone of the land plants in the latitude ~45°N–45°S and death of the tetrapod in 30°N–40°S is likely to be the response of the extreme El Niño effect after the PTME spanning in the low to middle latitudes". I realise this article is hot off the press and involves some authors here, but may be worth listing the other hypotheses of the trigger? By all means detail why you favour a particular mechanism. This may just be window dressing but worth taking the time to dedicate a couple of sentences to its interpretation and thought process behind it, rather than just throwing el nino's in randomly (which is how it comes across).

Line 196. Should "Biogeographic productivity" have a loose definition?

Line 246: Magnitude of predicted change?

Line 267. Silicate weathering "rate"? Or intensity?

Line 267: Would a better phrasing be: "than the direct effect of a decline in organic carbon burial"?

Line 281. If these mechanisms are "previously suggested" shouldn't there be some citations here?

Line 287. Suggest change to "extremely hot by Phanerozoic standards". Given this is quite an important sentence it could do with being a bit shorter and snappier I think.

288. Likewise "these conditions persisted for millions of years [can you place an estimate on this?], with cooling only achieved as plant productivity increased from the Middle Triassic".

Line 317 "cannot"

Line 486. Discussion of evolution of plant functional traits in these papers (if you like):

- Corenblit, D., Davies, N.S., Steiger, J., Gibling, M.R. and Bornette, G., 2015. Considering river structure and stability in the light of evolution: feedbacks between riparian vegetation and hydrogeomorphology. *Earth Surface Processes and Landforms*, 40(2), pp.189-207.
- Brückner, M.Z., McMahon, W.J. and Kleinhaus, M.G., 2021. Muddying the waters: modeling the effects of early land plants in paleozoic estuaries. *Palaios*, 36(5), pp.173-181.

Line 618. I'd be cautious about extending case study data which record local phenomena to an inferred widespread phenomena.

Figure 1. Can the font size be increased in this? Even if it ends up going in as a single page spread as it is at present it's quite hard to pick out the details. Perhaps uploads to the manuscript submission process is partly responsible.

Methodology: Should there not be details of the tetrapod source data in the methodology somewhere?

Reviewer #2

(Remarks to the Author)

I think this is a great paper. I have been a part of the movement towards putting paleobotanical information "to work" via modeling and this stands as a prime example of such. The modeling is in depth and complicated but well described. Every question that popped into my head was answered in the methods. All of the problems I found are superficial. I recommend this paper for publication.

Line 90–92: It took five tries to figure out what is being stated in this sentence. Please make more clear.

Throughout paper: Macrofossils, not Macro fossils. Plus, it's done both ways in the paper. Also, some paragraphs are indented, some not. There are little grammar mistakes throughout. I pointed some out, other I didn't. Please proofread, Line 212: figure reference is not correct.

Lines 276 – 278 I was just thinking about the significant effect of T on weathering. I'm glad you mentioned it.

Line 311: Tajikstan is mentioned twice.

Line 358 – 360: I'm not sure what this means. Representative of what? Local or regionally representation? Globally?

Line 522: There seems to be an unnecessary underscore symbol in this information.

Line 536: Cordaites is a species and should be capitalized, unless you meant the generic word "cordaitalean" which is mostly used in the surrounding sentences.

Reviewer #3

(Remarks to the Author)

Summary

The manuscript "Early Triassic super-greenhouse climate driven by vegetation collapse" suggests a new hypothesis for the 5-million-years long lasting greenhouse climate following the Permian-Triassic Extinction. To state this, the authors first use the fossil plant record to reconstruct the vegetation distribution and plant productivity before and after the mass extinction. Second, they use the reconstructed productivity for simulations with a climate-biogeochemical model to further investigate the details of a reduced plant productivity on climate. They interpret the results as an example for a threshold in the climate system above which the system stabilizes at a very warm climate for million of years.

General remarks

The hypothesis and the way it is investigated are very interesting and this paper is a relevant contribution for the better understanding of the Permian-Triassic extinction, but more generalized also of mechanisms in the Earth system.

However, I have several major remarks:

1. My largest difficulty with the manuscript is its structure which makes it hard for the reader to read and understand (and also for the reviewer!). The structure of the manuscript must be heavily revised. Please make clear where the description of what you do ends, where you start describing results and where you discuss and interpret the results. It might need more subheadings (if possible in the journal style?), or it might already work to make a section structure clearer by including blank lines for paragraphs or even by choosing a clearer wording.

One linked remark to this is the Methods section in the end. In my opinion most of it goes into a supplementary material in its current form. It needs substantial reduction and revision for a clear and short Methods part (see some ideas and details below).

2. As usually for deep time paleo studies, the fossil occurrences are very limited. This is the nature of exploring these time periods and in my opinion it is justified to still do it although the amount of data is not enough for a statistic investigation. So my critic is not that it is not enough data locations, but to discuss this problem more in depth and formulate conclusions based on very few data more cautiously. In particular I would like the authors to address or at least mention this issue for the tetrapod locations (for the Late Permian there is also not much data for the low latitudes, so please formulate your conclusions on the tetrapod distribution and in general the much higher low latitude extinction more carefully, l. 141-145). And for the extrapolation of the vegetation zones based on very few plant fossil locations. One particular question to discuss is the abundance of deserts in regions with low or no fossil occurrences. Why is it not just no preserved data? I think you touch this very shortly, but this part is not clear and too short.

3. Quality and selection of figures: the resolution of the figures is very low and together with a very small size this makes it difficult to read. In addition, instead of figure 1 I would prefer a figure which gives a broader overview of the changes. I think it could be reduced to less detailed information and diagram types which are intuitively understandable for an audience with a wider background. This is difficult for readers which do not have a botanical/paleobotanical background, but the topic and methods of the paper is relevant to a broader audience. Please also make sure in figure 1 and 3 that the axes and lines are clearly labeled and are not only understandable with the very long captions.

Some more detailed remarks:

Abstract: the very strong last sentence is difficult to understand: what exactly would be the link between the threshold idea and the warming?

Introduction: The first sentence of the second part (l.80-84) would be helpful to have in the introduction, so one knows what will be done.

Reconstructing plant productivity

l.95: it is not clear to me with this sentence and the referenced figure 1b why this is robust. Can you give more details here or refer to the Methods for this?

l.103: I was confused why you only mention here the South China Chanhshingian? Is it an example or the only duplicates or the highest?

l.134-136: this seems to be a very important sentence, but I only partly understand it and I am not sure on which part of the Methods you refer - can you be more specific or more precise where to find the information in the Methods?

l.145-147: I suggest to phrase this more carefully as I think it is very difficult and uncertain to establish a link like this

l. 148-161: I am not sure if this is something you conclude from your results or bring together with other results (as you cite so much other work) or confirm other results (if yes: what is new about your results/method)? Please make this clearer

l.189-192: something wrong with this sentence?

Modelling plant effects on long-term climate

l.208-221: is this necessary in this detail? Perhaps part of it could go to the Methods section

l.248/249: please make clear that the exceptions relate to the difference between the geological record and the fossil-NPP driven run. Would also be helpful to refer to figure 3 d and the blue and green line

l.254: mention that Sr is a weathering proxy for the broader audience?

l.255-257: is this shown somewhere? Like this I don't find this a helpful comparison

l.266-268: something wrong in the formulation? I do not fully understand

l.266-283: this seems to be a paragraph about the model limitations? I suggest to make it clear that you discuss the limitations and to make this part more concise and maybe also shorter

Materials and Methods: I suggest the largest part can go to a SI. I suggest also to check the headings as I find them not always fitting or at least not summarizing the topic of the section. I guess the model part would be good to stay in the Materials and Methods section.

additional remark on Figure 2: tetrapod dots should be smaller

Version 1:

Reviewer comments:

Reviewer #1

(Remarks to the Author)

The authors have done a great job responding to the reviewer comments.

(Remarks on code availability)

Reviewer #2

(Remarks to the Author)

My concerns have been addressed and I suggest publication.

(Remarks on code availability)

Reviewer #3

(Remarks to the Author)

I thank the authors for reading my review so carefully and for implementing many of my remarks. In particular, the paper is now much easier to read and understand due to a clearer structure.

I have only some minor remarks and recommend to accept it after these small changes (or at least the most relevant):

l.169-173: I guess modeling with HadCM3 was done in reference 9? This is not clear to me from the wording used. Could you make this clear by rephrasing it?

l.232 and following: it is not clear to me whether you need temperature fields for initialising the model (as you don't give radiation I would not know how to get at least the start temperature). If this is the case, where are the temperatures coming from?

l.250: add D for figure: (Fig. 4D)

l.258: (Fig.4E, G)

l.273: "(green line in Fig. 4)", please modify to "...Fig. 4C)"

l.280, add again where the data from the geological record is coming from

Figures:

Figure 2: this is a kind of diagram I am not so familiar with - if it is standard to do it the way it is, that's fine. I would prefer to have information of what is shown on the x-axis and also add latitude [°] for the y-axis.

Figure 3: it is cut off in the merged pdf version, but I guess the NPP maps are the same as in the first version of the manuscript and this is fine with me. I only have one remark to what is called NPPL in Figure 2 and Figure 3 as well as the text: I suggest to call it differently after CO₂ fertilization is included when the variable is used, for example like NPPL_f or similar

Figure 4: I find it confusing to use the same blue color (at least it looks like that) for the line in C and the line with points in D and G although the data in C is coming from a different data source. I suggest to change the color and also add a legend for this color. I would find it helpful to have in this legend already included where the recorded data is coming from for both data sources, so something like "recorded data from XX".

Figure S6: I expected that the 0.5 line should be the same as the green line in Figure 4G, but they seem to not agree with each other. Could you check this?

(Remarks on code availability)

I had a quick look at the code, but then realized it is only the general version of SCION. I guess the modified version is much more relevant (which they seem to provide as there is a sentence for this in the manuscript). In addition to the modified version, the authors need to make sure to provide a readme with clear instructions how to reproduce their results and the names of the variables they analyse and plot in their figures.

Response to reviews for Early Triassic super-greenhouse climate driven by vegetation collapse

Zhen Xu et al.

Reviewers' comments appear in black with our responses in blue. Modified text in the revised paper is also colored blue, with line numbers for new text noted here.

Reviewer #1:

Comment (1): Xu and colleagues shed light on the unusually long persistence of “super” greenhouse conditions following the Permian-Triassic mass extinction. Previous hypotheses for why it appears CO₂ could not be efficiently removed from the surface system include the reduced sediment flux of eroded material, which would limit silicate weathering, or contrastingly more rapid weathering regimes, both on the continents and in the oceans (reverse weathering). Neither satisfactorily explain why such conditions necessarily persist for 5 million years (approximately) after the event, and eventually recover during the middle Triassic. The authors counter this by proposing a model wherein the dramatic and prolonged diminishment of low-latitude plants takes centre stage. In the absence of tropical, peat-forming ecosystems substantially less CO₂ was likely to be sequestered from the end Permian. In this absence, the authors propose that Earth's climate-carbon cycle stabilized at a much higher average temperature than the perceived norm in the run up to the extinction. In this model, it would not be until these plant biomes sufficiently recovered that sustained cooling got momentum. I generally like this idea, and think it is very worthwhile adding it to the mix of possibilities. By putting this modelling-led work out into the public domain, various geochemical and petrological proxies might later be explored across a number of expanded sections, ultimately testing whether or not this new hypothesis is genuine. I see this as an important step in this scientific process. Subject to minor line-by-line comments, I recommend publication in Nature Communications

William McMahon

Response: We appreciate the reviewer's positive and detailed comments and have revised the manuscript to address them as outlined below.

Comment (2): Should Andrew Merdith's affiliation be updated?

Response: It should! We have done this. [line 12]

Comment (3): Line 63. I'm weary of such compilations of sedimentation rates honestly. How can one infer how much sedimentary stasis is locked up in individual bedding planes, and how this varies throughout an expanded section? Not as easy as just dividing thickness by time.

Response: We agree that sedimentation rates are uncertain, and had intended this to be part of our point here. We have revised the wording to specifically note that estimating sedimentation rates is difficult. [line 67]

Comment (4): Line 94. A few tens of thousands of years.

Response: Thanks for the correction, this section has been reworded significantly and moved to supplementary in line with other reviewer comments [line 1150]

Comment (5): Line 128. How does one link tetrapod occurrences to local climate? I'm not even entirely certain whether most paleobotany-climate links are so robust. Lithological indicators seem to be the key thing here (especially the three listed, in the absence of detailed accounts of sedimentary structures in other siliciclastic lithologies which may provide some clues). If we're talking tetrapod trackway's here, biases are more likely to pick out localized areas of greater accommodation/sediment aggradation rates, wherein the chances of reworking substrates where the trackways form decrease. So unless you have a huge amount of data to draw from, you'd maybe be picking out these preservational hot spots (or lack thereof), rather than anything meaningful for global climate. Also, is local climate really what's meant here. Or regional? Whilst I suspect the paleogeographic grid map is advanced, it's surely compartmentalized on a "regional" rather than "local" level?

Response: We regret being unclear in this section. The tetrapod fossils are used as an additional positive indicator of grid cells being occupied by vegetation, rather than an indicator of climate. We have made this clearer in the revision. We have also clarified the model grid size as both 'local' and 'regional' do not have well defined spatial scales. [line 147]

Comment (6): Line 134. Reference in incorrect place.

Response: Thanks for the correction, we have fixed this in the rewritten section.

Comment (7): Line 134. "The 'coal gap' after the PTME was not associated with a significant loss of river or delta sediments." Make it very clear how this is known. If it's not from a global database of sedimentary rock volume of a particular age and lithotype (see below for examples) I'd advise removing.

- Peters, S.E., Husson, J.M. and Czaplewski, J., 2018. Macrostrat: a platform for geological data integration and deep-time Earth crust research. *Geochemistry, Geophysics, Geosystems*, 19(4), pp.1393-1409)
 - McMahon, W.J. and Davies, N.S., 2018. Evolution of alluvial mudrock forced by early land plants. *Science*, 359(6379), pp.1022-1024.
 - Dunhill, A.M., Hannisdal, B. and Benton, M.J., 2014. Disentangling rock record bias and common-cause from redundancy in the British fossil record. *Nature communications*, 5(1), p.4818.
- I'd also suggest checking the references cited at this portion of the text (and maybe throughout) as a few seem out of place to me.

Response: We have rephrased this sentence to make it clear that we are discussing a series of local datasets in which the coal gap is observed but river or delta sediments continue in these sedimentary successions. We have made it clear that this is not a global database but we think it is worth including as it demonstrates how the coal gap is currently understood. The references for this in the main text and supplementary information have been checked and corrected where necessary. [line 155]

Comment (8): Line 146. "The extinction zone of the land plants in the latitude ~45°N–45°S and death of the tetrapod in 30°N–40°S is likely to be the response of the extreme El Niño effect after the PTME spanning in the low to middle latitudes". I realise this article is hot off the press and involves some authors here, but may be worth listing the other hypotheses of the trigger? By all means detail why you favour a particular mechanism. This may just be window dressing but worth taking the time to dedicate a couple of sentences to its interpretation and thought process behind it, rather than just throwing el nino's in randomly (which is how it comes across).

Response: We have rewritten this section to expand on this idea and to add a more comprehensive discussion of other extinction mechanisms [line 178].

Comment (9): Line 196. Should “Biogeographic productivity” have a loose definition?

Response: We modified this sentence to make it clearer that the ‘biogeographic productivity’ is a fossil-based reconstruction which does not consider the effects of CO₂ fertilization [line 244].

Comment (10): Line 246: Magnitude of predicted change?

Response: We were referring to the ‘fold change’ which is a ~2-4 fold increase in both our model and the proxy dataset. We have explicitly noted this now. [line 298]

Comment (11): Line 267. Silicate weathering “rate”? Or intensity?

Response: We have now noted that this is weathering intensity [line 320]

Comment (12): Line 267: Would a better phrasing be: “than the direct effect of a decline in organic carbon burial”?

Response: We agree. We have modified this sentence as suggested [line 321].

Comment (13): Line 281. If these mechanisms are “previously suggested” shouldn’t there be some citations here?

Response: We have added these citations [line 335].

Comment (14): Line 287. Suggest change to “extremely hot by Phanerozoic standards”. Given this is quite an important sentence it could do with being a bit shorter and snappier I think.

Response: We have made this change [line 343]

Comment (15): 288. Likewise “these conditions persisted for millions of years [can you place an estimate on this?], with cooling only achieved as plant productivity increased from the Middle Triassic”.

Response: We have now included an estimate here of nearly 5 million years [line 345]

Comment (16): Line 317 “cannot”

Response: We have modified this spelling mistake [line 367].

Comment (17): Line 486. Discussion of evolution of plant functional traits in these papers (if you like):

- Corenblit, D., Davies, N.S., Steiger, J., Gibling, M.R. and Bornette, G., 2015. Considering river structure and stability in the light of evolution: feedbacks between riparian vegetation and hydrogeomorphology. *Earth Surface Processes and Landforms*, 40(2), pp.189-207.
- Brückner, M.Z., McMahon, W.J. and Kleinans, M.G., 2021. Muddying the waters: modeling the effects of early land plants in paleozoic estuaries. *Palaios*, 36(5), pp.173-181.

Response: We have now incorporated these references [line 517].

Comment (18): Line 618. I'd be cautious about extending case study data which record local phenomena to an inferred widespread phenomena.

Response: We have modified this section to be more circumspect [line 652]

Comment (19): Figure 1. Can the font size be increased in this? Even if it ends up going in as a single page spread as it is at present it's quite hard to pick out the details. Perhaps uploads to the manuscript submission process is partly responsible.

Response: We have modified Figure 1 to make it easier to read and understand, and it is now supplied as a high-quality vector graphic.

Comment (20): Methodology: Should there not be details of the tetrapod source data in the methodology somewhere?

Response: We have now added the sources of data in the Figure 3 caption.

Reviewer #2:

Comment (1): I think this is a great paper. I have been a part of the movement towards putting paleobotanical information "to work" via modeling and this stands as a prime example of such. The modeling is in-depth and complicated but well described. Every question that popped into my head was answered in the methods. all of the problems I found are superficial. I recommend this paper for publication.

Response: We sincerely thank the reviewer for their kind and encouraging feedback. We have carefully addressed their detailed suggestions to improve the work.

Comment (2): Line 90–92: It took five tries to figure out what is being stated in this sentence. Please make more clear.

Responses: Our apologies. We have rewritten this part for clarity [line 100]

Comment (3): Throughout paper: Macrofossils, not Macro fossils. Plus, it's done both ways in the paper. Also, some paragraphs are indented, some not. There are little grammar mistakes throughout. I pointed some out, other I didn't. Please proofread,

Responses: We have modified this phrasing and have carefully proofread the revision.

Comment (4): Line 212: figure reference is not correct.

Responses: We have corrected this, thanks for spotting it. This text is now in the methods [line 696].

Comment (5): Lines 276 – 278 I was just thinking about the significant effect of T on weathering. I'm glad you mentioned it.

Responses: Thanks for the positive comment

Comment (6): Line 311: Tajikstan is mentioned twice.

Responses: This was a mistake in our abbreviations, so in the revised manuscript we use the full county names and have increased the font size to make this more legible in Figure 1 and S2.

Comment (7): Line 358 – 360: I'm not sure what this means. Representative of what? Local or regionally representation? Globally?

Responses: We have modified this wording to be clear that we have used all the published papers or datasets we could locate in our analysis [line 446]

Comment (8): Line 522: There seems to be an unnecessary underscore symbol in this information.

Responses: We have deleted this unnecessary symbol in the revision [line 550].

Comment (9): Line 536: Cordaites is a species and should be capitalized, unless you meant the generic word "cordaitalean" which is mostly used in the surrounding sentences.

Responses: Thank you for pointing out this mistake. We have revised the text to use "cordaitalean" as it refers to more than just the genus *Cordaites*. In the sentences specifically referring to the genus, we retained *Cordaites* and ensured it is italicized and capitalized.

Reviewer #3:

Summary

The manuscript "Early Triassic super-greenhouse climate driven by vegetation collapse" suggests a new hypothesis for the 5-million-years long lasting greenhouse climate following the Permian-Triassic Extinction. To state this, the authors first use the fossil plant record to reconstruct the vegetation distribution and plant productivity before and after the mass extinction. Second, they use the reconstructed productivity for simulations with a climate-biogeochemical model to further investigate the details of a reduced plant productivity on climate. They interpret the results as an example for a treshold in the climate system above which the system stabilizes at a very warm climate for million of years.

General remarks

The hypothesis and the way it is investigated are very interesting and this paper is a relevant contribution for the better understanding of the Permian-Triassic extinction, but more generalized also of mechanisms in the Earth system.

However, I have several major remarks:

Response: We thank the reviewer for their thoughtful summary and constructive evaluation of our manuscript. We appreciate their detailed feedback and have carefully considered the major remarks, as outlined below.

Comment (1): My largest difficulty with the manuscript is its structure which makes it hard for the reader to read and understand (and also for the reviewer!). The structure of the manuscript must be heavily revised. Please make clear where the description of what you do ends, where you start describing results and where you discuss and interpret the results. It might need more subheadings (if possible in the journal style?), or it might already work to make a section structure clearer by including blank lines for paragraphs or even by choosing a clearer wording.

One linked remark to this is the Methods section in the end. In my opinion most of it goes into a supplementary material in its current form. It needs substantial reduction and revision for a clear and short Methods part (see some ideas and details below).

Response: Thank you for this constructive point. We have modified the manuscript structure as best as we can within the journal guidelines and considering the combined reviewer comments. We have revised the headings to emphasize the main topics in each section, included clearer paragraph breaks, and have revised the Methods section for conciseness, with additional material now in the supplement.

Comment (2): As usually for deep time paleo studies, the fossil occurrences are very limited. This is the nature of exploring these time periods and in my opinion it is justified to still do it although the amount of data is not enough for a statistic investigation. So my critic is not that it is not enough data locations, but to discuss this problem more in depth and formulate conclusions based on very few data more cautiously. In particular I would like the authors to address or at least mention this issue for the tetrapod locations (for the Late Permian there is also not much data for the low latitudes, so please formulate your conclusions on the tetrapod distribution and in general the much higher low latitude extinction more carefully, l. 141-145). And for the extrapolation of the vegetation zones based on very few plant fossil locations. One particular question to discuss is the abundance of deserts in regions with low or no fossil occurrences. Why is it not just no preserved data? I think you touch this very shortly, but this part is not clear and too short.

Response: We agree with the reviewers point that more discussion is beneficial here. We have added a section to the paper noting the poor abundance of data, particularly in some locations and the need for us to infer vegetation from combinations of fossil and climate-sensitive lithological data. We note specifically that the tetrapod record is used here only to add gridcells where plants must have been abundant, rather than to restrict plant paleogeography where they are not found, and we have been clearer that the vegetation zones are based on climate proxies as well as fossils. Specifically, desert regions are denoted by lithological proxies indicating arid conditions, as well as climate model outputs that also suggest aridity [line 147, 167, 652]

Comment (3): Quality and selection of figures: the resolution of the figures is very low and together with a very small size this makes it difficult to read. In addition, instead of figure 1 I would prefer a figure which gives a broader overview of the changes. I think it could be reduced to less detailed information and diagram types which are intuitively understandable for an audience with a wider background. This is difficult for readers which do not have a botanical/paleobotanical background, but the topic and methods of the paper is relevant to a broader audience. Please also make sure in figure 1 and 3 that the axes and lines are clearly labeled and are not only understandable with the very long captions.

Response: We thank the reviewer for the constructive suggestions on the figures. We have expanded the information in Figure 1 into two figures to allow us to use full names and clearer diagrams. Figures are now also supplied as high quality vector graphics.

Some more detailed remarks:

Comment (4): Abstract: the very strong last sentence is difficult to understand: what exactly would be the link between the threshold idea and the warming?

Response: We have added a sentence on this to the main text which explains how beyond a certain temperature, vegetation die-back can result in further warming through removal of vegetation carbon sinks. We have also more clearly noted a reference to a recent numerical modelling paper specifically on this mechanism [346].

Comment (5): Introduction: The first sentence of the second part (1.80-84) would be helpful to have in the introduction, so one knows what will be done.

Response: We moved the first sentence to the end of the introduction as suggested [line 87].

Comment (6): Reconstructing plant productivity

1.95: it is not clear to me with this sentence and the referenced figure 1b why this is robust. Can you give more details here or refer to the Methods for this?

Response: We now number the methods section and refer this sentence to the corresponding part of the methods more clearly. We have also revised the wording here [line 105].

Comment (7): 1.103: I was confused why you only mention here the South China Chanhshingian? Is it an example or the only duplicates or the highest?

Response: Yes, it is just an example to show the impact of normalization. We modified this sentence to explain this more clearly [line 114].

Comment (8): 1.134-136: this seems to be a very important sentence, but I only partly understand it and I am not sure on which part of the Methods you refer - can you be more specific or more precise where to find the information in the Methods?

Response: We now make it clear that this refers to section 9 in the methods [line 155]

Comment (9): 1.145-147: I suggest to phrase this more carefully as I think it is very difficult and uncertain to establish a link like this

Responses: We have revised this section to be more balanced in line with this and other comments [line 178].

Comment (10): 1. 148-161: I am not sure if this is something you conclude from your results or bring together with other results (as you cite so much other work) or confirm other results (if yes: what is new about your results/method)? Please make this clearer

Response: We have modified this section to clarify that this is a new conclusion from our results, but is also consistent with previous studies that were either regional or were much more limited in scope [line 193].

Comment (11): 1.189-192: something wrong with this sentence?

Response: We have revised the grammar here, this was a mistake incorporating our own revisions - apologies for the confusion [line 237].

Comment (12): Modelling plant effects on long-term climate
1.208-221: is this necessary in this detail? Perhaps part of it could go to the Methods section

Response: We have removed as much of this as possible to the methods to provide a clearer overview for the reader [methods section 10]

Comment (13): 1.248/249: please make clear that the exceptions relate to the difference between the geological record and the fossil-NPP driven run. Would also be helpful to refer to figure 3 d and the blue and green line

Response: We have made these modifications as suggested [line 299].

Comment (14): 1.254: mention that Sr is a weathering proxy for the broader audience?

Response: We have made this useful addition [line 305].

Comment (15): 1.255-257: is this shown somewhere? Like this I don't find this a helpful comparison

Response: We have decided to remove this line as showing this more clearly would require further work to predict CIA directly from the model, which should be investigated in a separate study.

Comment (16): 1.266-268: something wrong in the formulation? I do not fully understand

Response: We have revised this section for clarity [line 320]

Comment (17): 1.266-283: this seems to be a paragraph about the model limitations? I suggest to make it clear that you discuss the limitations and to make this part more concise and maybe also shorter

Response: We now state these are limitations and shorten the section as much as we can [line 322]

Comment (18): Materials and Methods: I suggest the largest part can go to a SI. I suggest also to check the headings as I find them not always fitting or at least not summarizing the topic of the section. I guess the model part would be good to stay in the Materials and Methods section.

Response: We have moved parts of the methods into the SI as requested.

Comment (19): additional remark on Figure 2: tetrapod dots should be smaller

Response: We have done this in the revision.

Reviewer #1 (Remarks to the Author):

The authors have done a great job responding to the reviewer comments.

Reviewer #2 (Remarks to the Author):

My concerns have been addressed and I suggest publication.

Reply: We thank both Reviewer 1 and 2 for their former suggestions on our work.

Reviewer #3 (Remarks to the Author):

I thank the authors for reading my review so carefully and for implementing many of my remarks. In particular, the paper is now much easier to read and understand due to a clearer structure.

I have only some minor remarks and recommend to accept it after these small changes (or at least the most relevant):

Reply: We thank the reviewer 3 for all their detailed suggestions, which indeed helped improve this work. Updated line-by-line modifications in response to final comments are below.

l.169-173: I guess modeling with HadCM3 was done in reference 9? This is not clear to me from the wording used. Could you make this clear by rephrasing it?

Reply [Line 171]: Yes, the HadCM3 simulation was done in reference 9, Yadong Sun's 2024 science paper. We have added the reference 9 citation in this sentence for clarity.

l.232 and following: it is not clear to me whether you need temperature fields for initialising the model (as you don't give radiation I would not know how to get at least the start temperature). If this is the case, where are the temperatures coming from?

Reply [Line 251]: We have added a line on the model initiation. The model is given a starting CO₂ concentration and calculates temperature based on this and the solar flux.

l.250: add D for figure: (Fig. 4D)

Reply [Line 251]: Modified.

l.258: (Fig.4E, G)

Reply [Line 259]: Modified.

l.273: "(green line in Fig. 4)", please modify to "...Fig. 4C)"

Reply [Line 275]: Modified.

l.280, add again where the data from the geological record is coming from

Reply [Line 284]: Added citation

Figures:

Figure 2: this is a kind of diagram I am not so familiar with - if it is standard to do it the way it is, that's fine. I would prefer to have information of what is shown on the x-axis and also add latitude [°] for the y-axis.

Reply [Figure 2]: Both latitude and species number have now been added to Figure 2.

Figure 3: it is cut off in the merged pdf version, but I guess the NPP maps are the same as in the first version of the manuscript and this is fine with me. I only have one remark to what is called NPPL in Figure 2 and Figure 3 as well as the text: I suggest to call it differently after CO₂ fertilization is included when the variable is used, for example like NPPL_f or similar

Reply [Throughout paper]: Good suggestion. We have modified the NPPL into NPPL_f in the main text. In the figure 2, we only show the original NPPL because the CO₂ fertilization effect is variable within the model run.

Figure 4: I find it confusing to use the same blue color (at least it looks like that) for the line in C and the line with points in D and G although the data in C is coming from a different data source. I suggest to change the color and also add a legend for this color. I would find it helpful to have in this legend already included where the recorded data is coming from for both data sources, so something like "recorded data from XX".

Reply [Figure 4]: We prefer the consistency of having the model and geological record in consistent colours throughout, which is how SCION papers have typically been displayed. We have checked the caption to ensure all data sources are included.

Figure S6: I expected that the 0.5 line should be the same as the green line in Figure 4G, but they seem to not agree with each other. Could you check this?

Reply [Figure S6]: Thanks for spotting this. Figure S6 was from an older version of the paper and the lines were shifted slightly. We have updated this now. All text is unaffected.

Reviewer #3 (Remarks on code availability):

I had a quick look at the code, but then realized it is only the general version of SCION. I guess the modified version is much more relevant (which they seem to provide as there is a sentence for this in the manuscript). In addition to the modified version, the authors need to make sure to provide a readme with clear instructions how to reproduce their results and the names of the variables they analyse and plot in their figures.

Reply [Lines 411–413]: We have deposited the modified version of the code and all the forcing/data files in Zhen Xu's Github. Zhen will make it open access once the paper is online. Here is the Github link: https://github.com/ZhenXuJane/SCION_Xu2025 A ReadMe file is also listed there to show how to repeat the code.

I think this is a great paper. I have been a part of the movement towards putting paleobotanical information “to work” via modeling and this stands as a prime example of such. The modeling is in depth and complicated but well described. Every question that popped into my head was answered in the methods. My of the problems I found are superficial. I recommend this paper for publication.

Line 90–92: It took five tries to figure out what is being stated in this sentence. Please make more clear.

Throughout paper: Macrofossils, not Macro fossils. Plus, it’s done both ways in the paper. Also, some paragraphs are indented, some not. There are little grammar mistakes throughout. I pointed some out, other I didn’t. Please proofread,

Line 212: figure reference is not correct.

Lines 276 – 278 I was just thinking about the significant effect of T on weathering. I’m glad you mentioned it.

Line 311: *Tajikstan* is mentioned twice.

Line 358 – 360: I’m not sure what this means. Representative of what? Local or regionally representation? Globally?

Line 522: There seems to be an unnecessary underscore symbol in this information.

Line 536: *Cordaites* is a species and should be capitalized, unless you meant the generic word “cordaitalean” which is mostly used in the surrounding sentences.